# Effect of Different Coupling Agents on Interfacial Properties of Fibre-Reinforced Aluminum Laminates

**DOI:** 10.3390/ma14041019

**Published:** 2021-02-21

**Authors:** Wei Zhu, Hong Xiao, Jian Wang, Xiudong Li

**Affiliations:** 1National Engineering Research Center for Equipment and Technology of Cold Strip Rolling, Yanshan University, Qinhuangdao 066004, China; zwdlf@stumail.ysu.edu.cn (W.Z.); xhh@ysu.edu.cn (H.X.); l18733521396@163.com (X.L.); 2State Key Laboratory of Advanced Design and Manufacturing for Vehicle Body, Hunan University, Changsha 410082, China

**Keywords:** fibre-metal laminates (FMLs), low-velocity impact, surface modification, XPS analysis, energy absorption

## Abstract

Metal composite interface properties significantly affect the integrity, bonding properties, and interface structure of Fibre Metal Laminates (FMLs). Interfacial bonding strength’s effect on Carbon Fibre-Reinforced Aluminium Laminate (CARALL) mechanical behaviours was investigated via three-point bending and low-velocity impact tests. AA6061 sheets were subjected to surface pretreatments under three conditions (anodizing and A-187 and A-1387 surface modifications) to obtain different interfacial bonding strengths. The bonding interfaces of CARALL were analysed using scanning electron microscopy, energy dispersive spectroscopy and X-ray photoelectron spectroscopy. Interfacial bonding strength between aluminium alloy and epoxy resin was determined by the tension-shear test. CARALL’s energy absorption capacity and failure mode were analysed after low-velocity impact and three-point bending under different aluminium alloy volume contents and surface pretreatments. Upon modification of metal surfaces, the interfacial bonding strength increased, and the highest was obtained by silane coupling agent A-1387. Improved strength maintained FML’s integrity under quasi-static and dynamic loadings. A-1387 improved the bonding ability of aluminium alloy and Carbon Fibre-Reinforced Plastics (CFRP). The composite interface strongly resisted crack propagation because of its functional group characteristics. When the volume content of aluminium alloy was less and greater than that of CFRP, the energy absorption capacity of CARALL weakened and strengthened, respectively, with increasing interfacial bonding strength.

## 1. Introduction

Compared with metal materials, fibre metal laminates (FMLs) have excellent fatigue resistance. After the generation of microcracks in metal components, crack growth and propagation speed become extremely fast, causing sudden fracture failure in aircraft operation. The crack only propagates in the metal layer due to the crack bridging mechanism of FMLs, and the adjacent fibre layer plays a role in bonding and bridging to inhibit crack growth. Therefore, FMLs have good damage tolerance and fatigue performance. Even if the FMLs appear to be cracked, they still have sufficient strength, which greatly extends the service life of materials under load. An aircraft will encounter extreme environments during operation, such as lightning strikes, hail, bird impact and debris impact on the orbit [1,2,3]. Thus, the fuselage material should have good impact and corrosion resistance. FMLs can effectively solve these problems [4,5,6,7,8,9].

Lightweight FMLs have received widespread attention due to the shortcomings of common aluminium alloys. In some cases, for example, anti-fatigue cracks cannot withstand high pressure. With the deepening of research in recent years, a series of FMLs with better performance had been developed [5,10,11]. These materials have excellent mechanical and physical properties [5,12,13,14]. Among them, Carbon Fibre-Reinforced Aluminium Laminates (CARALL), such as aramid-reinforced aluminium laminates and glass-reinforced aluminium laminates, are better than other Al-based FMLs in terms of strength, stiffness and fatigue performance [15]. Carbon fibre (CF) has higher modulus and residual and impact strength and lower specific strength than aramid fibre and glass fibre. Moreover, compared with aramid and glass fibres, CF can provide a more effective crack bridging to Al layers, thereby effectively reducing crack occurrence [16,17]. The crack bridging mechanism of FMLs is shown in Figure 1 [18]. However, the adhesion between aluminium alloy and the CF layer is relatively weak and needs improvement [19].

The stiffness and failure mode of composites can be strongly influenced by metal composite interface, although it does not consume much energy in the failure process of FMLs. Generally, two methods, namely, adhesive modification and the surface pretreatment of metal sheet, are widely used to improve the interface properties between carbon fibre-reinforced plastics (CFRP) and metal. Compared with the improvement of the proper adhesive’s design, metal surface pretreatment is an effective way to enhance the interfacial bonding strength between CFRP and metal.

Zhai et al. [20] reported the effects of anodizing voltage and time on the tensile strength and interlaminar shear strength of FMLs and discussed the impact of aluminium alloy/resin interface on the mechanical properties of FMLs. Zhang et al. [21] showed that the modification of the metal surface can make the interface layer realise the chemical bonding mode, which greatly improves the shear strength. Mei et al. [22] introduced one dendrimer layer into the epoxy/metal interface to effectively transfer the load from the epoxy resin to the metal surface, thereby greatly improving the bonding strength of the interface. Hamill et al. [23] studied the use of bulk metallic glass instead of aluminium alloy to improve bond strength. Zhang X. et al. [24] introduced multi-walled carbon nanotubes at the interface between metal and CFRP to improve the interfacial properties of FMLs. Although several studies have been conducted [25,26] on the preparation technology of FMLs, the weak wettability between aluminium alloy and CFRP is the main cause of the low interfacial bonding strength and the decline of mechanical properties of FMLs. The present study aimed to achieve excellent interfacial bonding strength between aluminium alloy and CFRP to improve mechanical performance.

Three surface-pretreatment methods, namely, anodizing, epoxy and amine silane coupling agents, were used to prepare sandwich Al/CFRP/Al laminates. The effects of different functional groups on the micro interface structure and macro mechanical properties were estimated in detail. The interface bonding strength were characterised by the tension–shear test. The interfacial bonding mechanism was analysed by scanning electron microscopy (SEM) and X-ray photoelectron spectroscopy (XPS). Finally, the effects of interfacial bonding strength and aluminium alloy volume content on the mechanical behaviour of CARALL were investigated by three-point bending test and low-velocity impact test. The deformation failure mode and energy absorption capacity of CARALL were analysed and discussed.

## 2. Experiment

### 2.1. Materials

The T700 CFRP unidirectional prepreg with 0.1 mm thickness and 40% epoxy resin content was provided by Zhongfu Shenying Carbon Fibre Co., Ltd. (Lianyungang, China). The AA6061-T6 sheet was used as the metal material. The geometric size of AA6061 sheets was 130 mm × 120 mm. The elemental constituents of 6061 are given in Table 1. The thicknesses were 0.4, 0.6 and 0.8 mm. The epoxy type silane coupling agent A-187 and the amine type silane coupling agent A-1387 were supplied by Silquest (Friendly, WV, USA). The chemical names and structural formulas are given in Table 2.

### 2.2. Electrochemical Treatment of AA6061 Sheets

AA6061 sheets were subjected to surface pretreatments under three conditions, namely, anodizing and A-187 and A-1387 surface modifications, and the appropriate sequences are presented in Table 3. At the beginning of the experiment, the phosphoric acid anodizing (PAA) process was conducted using a grinding machine (Songqi Co., Ltd., Qinhuangdao, China) equipped with 180# SiC papers. AA6061 sheets were first cleaned in sodium hydroxide solution (NaOH (Songqi Co., Ltd., Qinhuangdao, China), 100 g/L) and then in nitric acid solution (HNO_3_ (Songqi Co., Ltd., Qinhuangdao, China), 100 g/L) to remove the oxides and impurities on the surface. To further clean AA6061, the deionised water was used to thoroughly rinse the surface. Afterward, the AA6061 sheets were immersed in phosphoric acid (H_3_PO_4_, 100 g/L) electrolyte, where the sheets were selected as the anode. The voltage of DC power, anodizing temperature and anodizing time were 25 V, 25 ± 2 °C and 20 min, respectively. Finally, the AA6061 sheets were taken out for cleaning with deionised water.

In the 187 and 1387 processes, the AA6061 sheets were immersed in a hydrolysis solution of A-187 silane and A-1387 silane for 5 min after A pretreatment, respectively. The silane hydrolysis solution was hydrolysed by 5:5:90 volume ratio of silane coupling agent, deionised water and ethanol. The operation steps were as follows: add the mixed solution of absolute ethanol and deionised water into the beaker; stir and drop the acetic acid; adjust the pH value to 4; and perform ultrasonic oscillation for 30 min. Finally, the AA6061 sheets were dried with cold wind. The hydrolysis and condensation reactions of typical organofunctional silanes were as follows [27,28,29,30]:

(1) Hydrolysis:
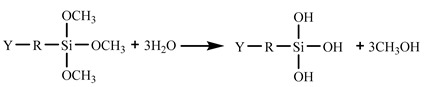
(1)

Y: reactive functional groups, R: molecular chain.

(2) Condensation with silanol on metal surface (adhesion/coupling).

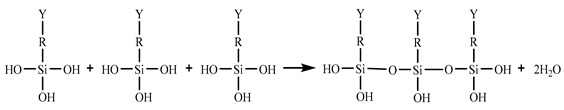
(2)

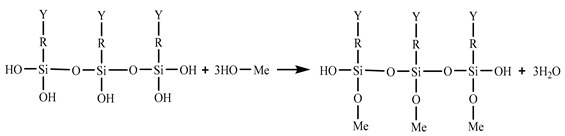
(3)

(3) Condensation with another silane (crosslinking).

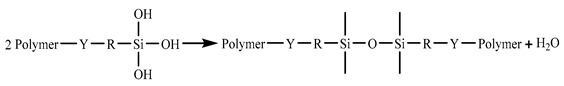
(4)

### 2.3. Fabrication of FMLs

Three categories of CARALL with pattern of 2/1 FMLs, namely, two AA6061 sheets and one CFRP ply, were prepared and their laminating designs were shown in Figure 2a. The curing process was conducted in an autoclave (Meizhoubao special equipment Co., Ltd., Taizhou, China) with optimised parameters, as shown in Figure 2b. Primarily, the temperature of the autoclave was increased to 80 °C with a heating rate of 2 °C/min and held for 30 min to ensure that the thermosetting resin contributed to adequate fluidity and air removal. Then, the temperature was increased to 115 °C with the same heating rate and kept for 60 min to cure the epoxy resin at the interface first. Thereafter, the samples were further subjected to an increase in temperature (145 °C), which was maintained for 120 min to cure the epoxy resin. Finally, the specimens were cooled to 25 °C to obtain the cured CARALL, as shown in Figure 2c. The prepreg lay-up method and aluminium sheet thickness in each CARALL sample are shown in Table 4.

### 2.4. Experimental Setup

Three mechanical tests were carried out in this investigation, namely, tension–shear, three-point bending and low-velocity impact tests. These tests are discussed in detail as follows.

#### 2.4.1. Tension-Shear Test

The specimens for the tension-shear test were made according to the standards of GB/T 6396-2008 (Clad steel plates–mechanical and technological test [31]). Tension shear tests of PAA-S, 187-S and 1387-S were carried out. The tensile rate was 1 mm/min. A certain thickness of the reinforcing sheet was bonded on the aluminium side close to the shear surface to ensure that the shear plane and tensile force were in the same plane. Four samples from each condition were tested, and the average values with standard deviations were given. The schematic of the tension–shear test specimen is shown in Figure 3.

#### 2.4.2. Three-Point Bending Test

The bending strengths of CARALL were determined according to ASTM Standard D7264 [32]. The loading speed was 1 mm/min, and the support span-to-thickness ratio was 32:1. Four samples from each condition were tested, and the average values with standard deviations were given.

#### 2.4.3. Low-Velocity Impact Test of CARALL

A low-velocity impact test was carried out on 100 mm × 100 mm specimens by using an Instron 9350 drop-weight test machine (Instron Corporation, Canton, MA, USA) that can record load-time history. A hemispherical impactor tip with a diameter of 12.7 mm was used, and the whole impact mass was 5.131 kg. All the conducted low-velocity impact testes were based on the ASTM D7136 standard [33]. In the current test, the impact velocity was 3.42 m/s.

### 2.5. Interface Analysis Methods

Samples of microstructure examination were taken from the thickness coordinate. Before conducting SEM (ZEISS Sigma 500, Carl Zeiss AG, Hallbergmoos, Bayern, Germany), the indispensable step of polishing the surfaces of the samples was performed. For SEM characterisation, the extra high tension of the SEM was 15.0 kV, and the detector was SE2 (Carl Zeiss AG, Hallbergmoos, Bayern, Germany). Furthermore, by the application of the SEM and energy dispersive X-ray spectroscopy (EDS), the distribution of elements on the interface was studied. To examine the element composition and the chemical form of interface, X-ray photoelectron spectroscopy (XPS) analysis was carried out on a Thermo Scientific K-Alpha spectrometer (Thermo Fisher Scientific, Waltham, MA, USA), equipped with an ALK alpha source (1486.8 eV photon) and an X-ray spot size of 500 μm.

## 3. Results and Discussion

The surface pretreatment processes for AA6061 sheet were sanding method, corrosion method, anodizing and addition of silane coupling agent and fabrication of CARALL. These methods are shown schematically in Figure 4.

### 3.1. Characterization of Pretreated AA6061 Sheet

To improve the interface performance of CARALL, a flexible silane layer was prepared on the anodized AA6061 sheet. Aluminium was anodized in phosphoric acid solution. The formation process of anodized film and the dissolution process of aluminium anodized film were opposite and closely related to each other. The formation of alumina film and the dissolution of alumina occurred simultaneously at the anode.

Film formation process:(5)$$2Al+3H_{2}O\;\to Al_{2}O_{3}+6H^{+}+6e$$

Dissolution process:(6)$$Al_{2}O_{3}+\;6H^{+}\;\to 2Al^{3+}+\;3H_{2}O$$

The decomposition of water occurred on the cathode and hydrogen was released:(7)$$6H_{2}O+6e\;\to \;3H_{2}↑\;+\;6OH^{-}$$

In phosphoric acid solution, oxide formation and dissolution are not simple reactions. The anion ${PO}_{4}^{3-}$ participates in the cathodic reaction process of aluminum and finally forms an anodized film (Al_2_O_3_·Al(OH)_x_(PO_4_)_y_) containing phosphate radical. The reaction process is as follows:(8)$$Al^{3+}+\;xH_{2}O+\;yPO_{4}^{3-}\;\to Al{\left(OH\right)}_{x}{\left(PO_{4}\right)}_{y}+\;xH^{+}$$

Therefore, several hydroxyl groups are present in the anodized film, which reacts with the ring-opening epoxy chain to form Al–O–C bond during the curing process and form Al–O–Si chemical bond by a condensation reaction with the silanol group of coupling agents [34,35,36].

Figure 5 shows the SEM image of the surface of AA6061 after pretreatment. On the surface of the pretreated phosphoric acid anodic oxide film, the distribution of nanopores was very regular, and the pore size was uniform. Moreover, a certain surface dendritic structure was observed on the anodized film.

Figure 6 shows the cross-section of CARALL in processes PAA, 187 and 1387. No obvious gaps or other defects were observed under SEM. Specifically, no transition trace existed at the interface layer under 187 and 1387 processes, which indicated that an excellent wettability and adsorption effect was achieved. An interface layer with a thickness of approximately 2 μm was formed at the bonding interface of aluminium alloy and CFRP, and this interface layer was tightly combined, as shown in Figure 7a and Figure 8a. A transition layer was produced at the interface between CFRP and AA6061 with decreasing concentrations of Al and C and with increasing contents of the O element, as shown in Figure 7b–d and Figure 8b–d. The surface modification pretreatment produced a specific flexible interface layer, which could relieve and release the internal stress caused by temperature during the curing process [37,38].

To study the bonding behaviour of the surface pretreatment in the interface layer, the chemical state of the interface layer of the CARALL plate was studied via XPS. The XPS spectra of C1s, Al2p and O1s obtained from the bonding interface of CFRP and AA6061 after PAA pretreatment process are shown in Figure 9. The C1s spectra detected from the bonding interface can be resolved into three component peaks, namely, C–C/CH, C–O and C=O, corresponding to 284.8, 285.83 and 288.0 eV, respectively. The Al2p peak can be resolved into three distinct component peaks, namely, oxide part of Al_2_O_3_, metal Al and coupling agent and metal surface hydroxyl reflecting the production part of Al-O-Si, which corresponded to 75.4, 73.1 and 72.1 eV, respectively. At the same time, the O1s peak can be resolved into four distinct component peaks, namely, Al_2_O_3_, C–O, C=O and C–O–A, which corresponded to 532.1, 533.2, 531.2 and 530.9 eV, respectively. The appearance of the C–O–Al bond indicated that the hydroxyl group in the alumina chemically reacted with the epoxy group in the resin.

The XPS spectra of C1s, Al2p and O1s at the bonding interface of CFRP and AA6061 with 187 and 1387 pretreatment processes are shown in Figure 10 and Figure 11, respectively. Unlike the unmodified interface, the C1s spectra detected from the interface can be resolved into five different peaks, namely, C–C/CH, C–O, C=O and two new peaks of C–Si and C–N corresponding to 284.2 and 285.6 eV, respectively. However, the Al2p spectra of the two bonding interfaces can also be resolved into three different peaks, namely, Al_2_O_3_, Al and Al–O. In addition, the O1s peak in Figs. 10 and 11 are resolved into five different peaks, namely, Al_2_O_3_, C=O, C–O, O–Al (binding energy of 531.1 eV) and a new Si–O peak (binding energy of 532.5 eV). The existence of Si–O–Al chemical bonds demonstrated the presence of coupling agents and the successful modification of the AA6061 surface. The existence of the C–O–Al chemical bond proved that anodizing caused a chemical reaction between the resin and the alumina. The existence of C–Si, Si–O–Al and C–N chemical bonds demonstrated the presence of coupling agents and the successful modification of the AA6061 surface.

Figure 5 shows that the anodizing pretreatment makes the aluminium alloy surface form a specific nanostructure anodized film. The anodized film increases the wettability of resin and aluminium alloy surface and promotes the interface chemical reaction because of the large amount of hydroxyl groups in the anodized film. Figure 7, Figure 8, Figure 10 and Figure 11 depict that addition of coupling agent flexible layer greatly increased wet adhesion. The silane coupling agent molecules formed a flexible layer at the interface between aluminium alloy and resin, which can effectively and evenly transfer load and reduce stress concentration at the interface. Therefore, the introduction of coupling agent at the interface can greatly increase the bonding strength of the interface and the entirety of the mechanical properties of CARALL.

### 3.2. Interfacial Bonding Property of CARALL

The interlaminar shear strength test is a method used to measure the interlaminar characteristics of CARALL. It reflects the strength of the interface between fibre and metal in FMLs from the perspective of the application. In this research, the interfacial bonding strength of CARALL with different AA6061 surface pretreatments was tested by the tension–shear method, as shown in Figure 12.

A flexible coupling agent layer was prepared on the anodized AA6061 sheet to improve the bonding durability of CARALL. The mechanism underlying the enhancing effect of A-187 and A-1387 on the interfacial strength is shown in Figure 4. The inorganic functional groups of the two silane coupling agents were trialkoxysilyl groups, namely, Si–(OCH_3_)_3_. The trialkoxysilyl group hydrolysed to silane alcohol and then condensed with hydroxyl groups on the surface of the AA6061 sheet to form a chemical bond, as shown in Figure 4. The organic functional groups were involved in the curing reaction of epoxy resin matrix. The epoxy group at the end of the molecular chain of A-1387 reacted with the molecular chain of amine curing agent in the resin system, which effectively connected the aluminium alloy with the resin and the network structure. The coupling agent molecules formed at the interface between the aluminium sheet and the resin, which effectively and evenly transferred the load and reduced the interfacial stress concentration. Therefore, the introduction of the molecular chain at the interface can effectively increase the bonding strength of the interface. The interfacial shear strength increased from (49.5 ± 1.3) MPa to (53.4 ± 1.0) MPa.

After two weeks, the percentage of interfacial shear strength decreased from 10.5% to 2.6%. In the same way, the end of the molecular chain of A-1387 also had a reactive primary amino group containing two active hydrogen atoms that can bond with the main epoxy resin molecular chain. Thus, the interfacial bond strength improved with the addition of the coupling agent. On the one hand, compared with the epoxy resin coupling agent, the amine coupling agent had more action points with the main network structure of the epoxy resin. On the other hand, ether bonds (–C–O–C–) were present in the organic molecular chain of the epoxy coupling agent, and the fracture energy was slightly lower than that of alkyl (–C–C–). The molecular chain strength at the interface layer was lower, which more easily led to fracture. Therefore, the interfacial shear strength of the system with amine coupling agent A-1387 was higher than that of A-187, which is (58.3 ± 1.1) MPa. After two weeks, the interfacial shear strength decreased only by 0.9%. The bonding strength without any treatment was only about 7 MPa [39]. In addition, Figure 12 shows that the primary amino group also existed in the molecular chain of A-1387; thus, it reacted with the epoxy molecular chain. Cross-linking between the molecular chains effectively increased the cohesive strength and modulus of the interface layer, which was beneficial to the interface load transfer and stress concentration reduction [40]. Therefore, the interfacial shear strength of the A-1387 system was relatively high.

### 3.3. Bonding Property of CARALL

Figure 13 shows the effect of different surface pretreatment methods (PAA, 187 and 1387) on the bending strength of CARALL, where the thickness values of the AA6061 sheet were 0.4, 0.6 and 0.8 mm. The bending strength of CARALL after three surface pretreatments was consistent with the interfacial shear strength obtained by tension–shear test. With increasing interfacial bonding strength, the bending strength also increased, which further verified that the surface modification of amine coupling agent A-1387 had the greatest effect on the improvement of the interfacial bonding between AA6061 sheet and CFRP. In addition, the volume content of the AA6061 sheet in CARALL also affected the bending strength. The strength decreased with increasing AA6061 sheet volume content. The A-1387-modified CARALL exhibited excellent flexural properties in the bonding strength test, because the molecular chain of A-1387 was relatively long, which inhibited the formation rate of Griffith microcracks that reduced CFRP strength during CARALL‘s slow bending process. As shown in Figure 14, in the bending process, the cracks first appeared in the interface layer far away from the stress point, and then, the cracks initiated and propagated upward. Therefore, with an increasing number of CFRP layers, the energy required for crack propagation increased. Under the same thickness of the CARALL plate, the bending resistance increased with decreasing AA6061 sheet thickness, consistent with the results presented in Figure 13.

### 3.4. Low Velocity Impact of CARALL

To determine the impact of the surface pretreatment on the low-velocity impact response of CARALL, different volume contents of AA6061 sheet in CARALL were predicted and tested by using a standard drop-weight test machine, as shown in Table 5.

The interface bonding strength had no significant effect when the impact velocity was low, and metal fracture was absent [41]. Therefore, the effect of AA6061 sheet volume content and interface bonding strength on the deformation/failure mode of CARALL was studied by using 3.42 m/s impact velocity to completely penetrate CARALL. The failure modes of the front and back panels of the three kinds of CARALL were the same. The front panel was circular, and the back panel caused star-shaped cracks under dynamic load, thereby forming petal failure mode, as shown in Figure 15.

The displacement force curves and energy absorption of the nine types of CARALL are compared in Figure 16. When the volume contents of aluminium alloy were the same, and the strengths of interface differed, the initial loading processes of CARALL were nearly the same, as indicated in Figure 16a–c. After the initial loading stage, debonding occurred, and the integrity of CARALL was compromised. When the volume content of AA6061 was less than that of CFRP, the energy absorption decreased with increasing interface bonding strength, because the CFRP delamination and fibre shear fracture energy absorption dominated. The increase of interface bonding strength reduced the failure area of CARALL, thereby hindering the energy absorption of CARALL. When the volume content of AA6061 was larger than that of CFRP, the energy absorption of CARALL increased with increasing interface bonding strength. This finding was due to the fact that the plastic deformation energy absorption of AA6061 was dominant, and the increase of interface bonding strength increased the energy requirement of the plastic deformation of AA6061, thereby increasing CARALL’s energy absorption. Therefore, the C-0.8 sample showed the best specific energy absorption. Figure 16d–f show that when the aluminium alloy volume contents were different, and the interface bonding strengths were the same, the trend of the whole loading process of CARALL was basically the same. With increasing aluminium alloy volume content, the force-displacement curve moved backward, showing that the plastic deformation energy absorption gradually dominated.

## 4. Conclusions

The AA6061 surface was processed to produce homogeneous nanoporous structure after anodizing. This structural component increases the mechanical interlock effect of CARALL. Chemical bonding between epoxy resin and anodized film can occur because of the abundant hydroxyl groups in the anodized film.

Surface modification of silane coupling agents A-187 and 1387 after anodizing can improve the interfacial bonding strength and bending strength of CARALL and enhance the performance degradation of the composite interface layer prepared by hot pressing. After placing the CARALL for 2 weeks, the shear strength of the modified epoxy coupling agent decreased by 2.6% compared with that of the unmodified one (12%), whereas the shear strength of the amine coupling agent only decreased by 0.9%. This finding shows that the addition of the flexible layer greatly improves the stability of CARALL’s mechanical properties, and the amine coupling agent exhibits excellent performance.

In the low-velocity impact test, the influence of interfacial bonding strength on CARALL energy absorption is related to the aluminium alloy volume content. When the volume content of AA6061 is smaller than that of CFRP, the energy absorption decreases with increasing interfacial bonding strength. When the volume content of AA6061 is greater than that of CFRP, the energy absorption of CARALL increases with increasing interfacial bonding strength.

The surface modification of amine coupling agent A-1387 enables CARALL to exhibit excellent tensile strength and flexural strength. On the one hand, many interaction points exist between the amine coupling agent and main network structure of the epoxy resin. Thus, cross-linking can occur between molecular chains. This linking can lead to a relatively high bonding strength and modulus of the interface layer, which benefits the interface load transfer and stress concentration reduction. On the other hand, the fracture energy of alkyl in amine coupling agent is slightly higher than that of ether bond in epoxy coupling agent. Therefore, the amine coupling agent system can obtain higher interfacial shear strength. The macromolecular chain of the amine coupling agent can effectively protect CFRP from forming Griffith flaw, namely, decreasing strength.

## Figures and Tables

**Figure 1 materials-14-01019-f001:**
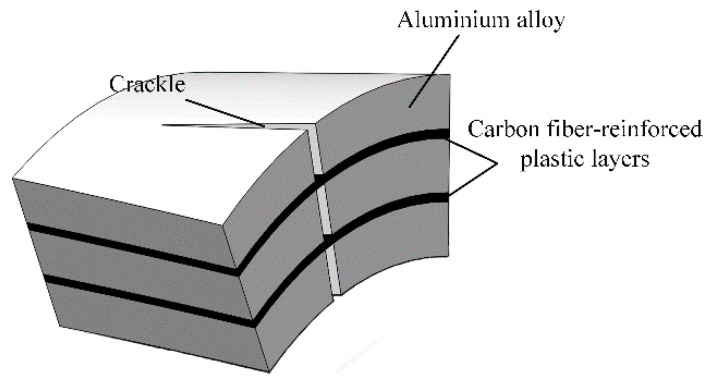
Mechanism of crack bridging.

**Figure 2 materials-14-01019-f002:**
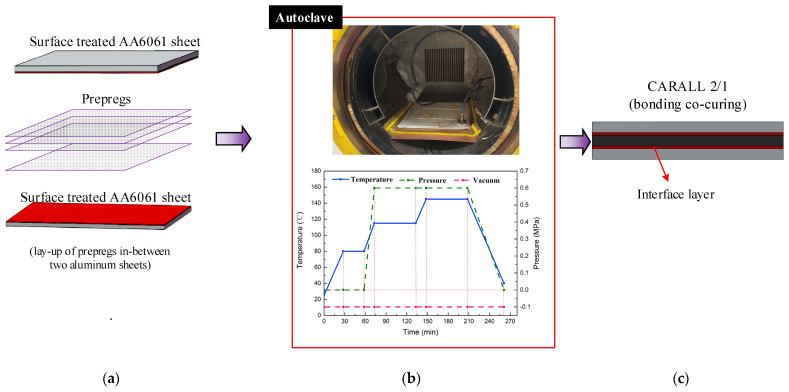
Fabrication flow process of workpiece: (**a**) Hand lay-up, (**b**) Curing process, (**c**) Samples.

**Figure 3 materials-14-01019-f003:**
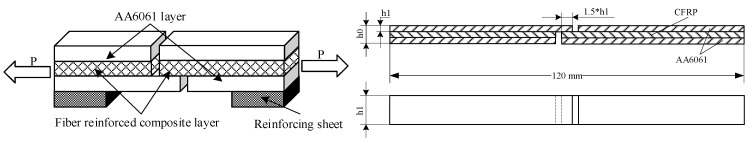
Tension-shear test of CARALL plate.

**Figure 4 materials-14-01019-f004:**
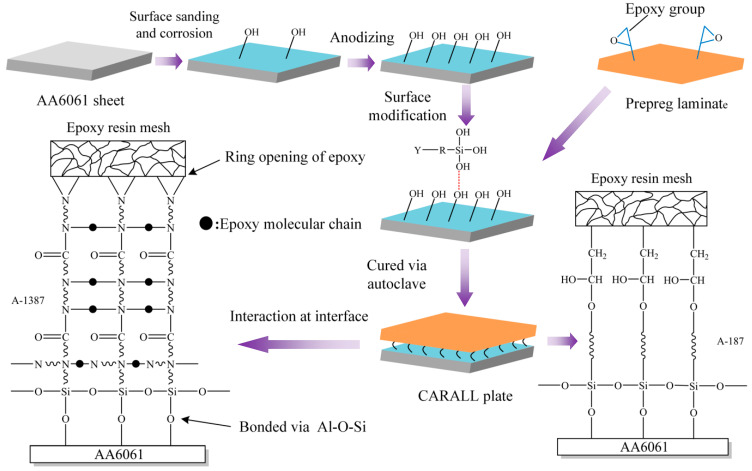
Schematic representation of CARALL processes.

**Figure 5 materials-14-01019-f005:**
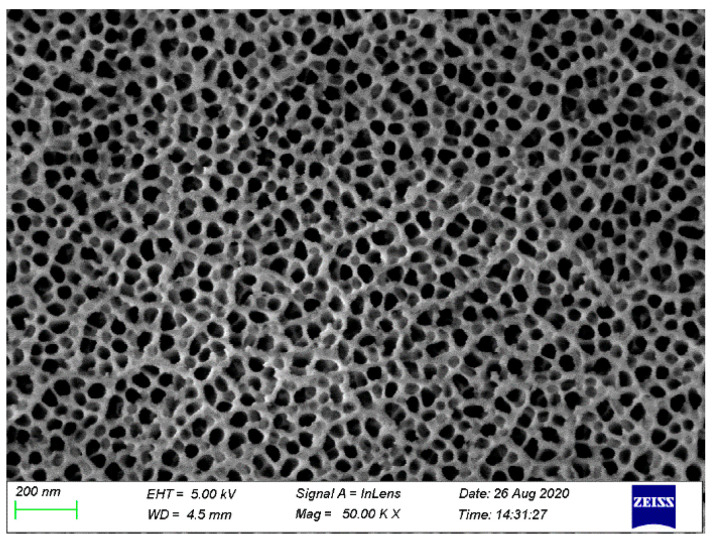
Surface topography of AA6061 anodized at 25 V for 20 min.

**Figure 6 materials-14-01019-f006:**
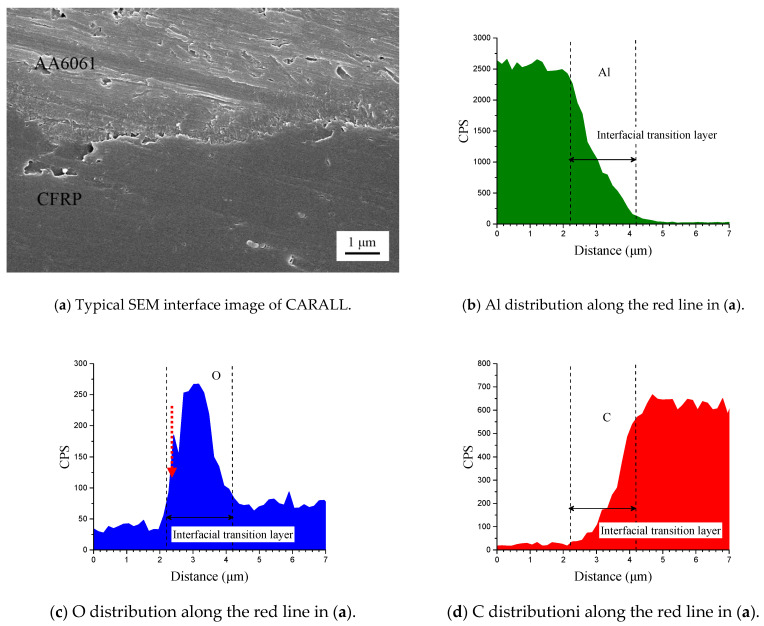
Typical bonding interface of specimen PAA.

**Figure 7 materials-14-01019-f007:**
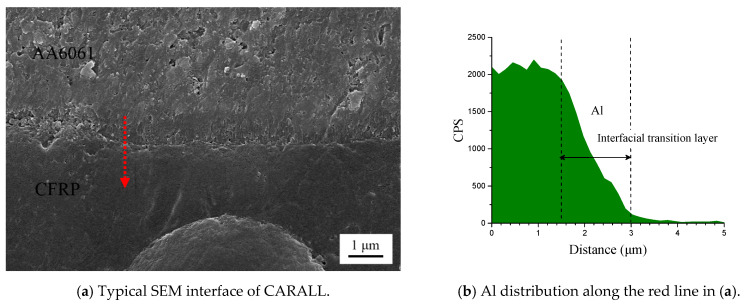
Typical bonding interface of specimen 187.

**Figure 8 materials-14-01019-f008:**
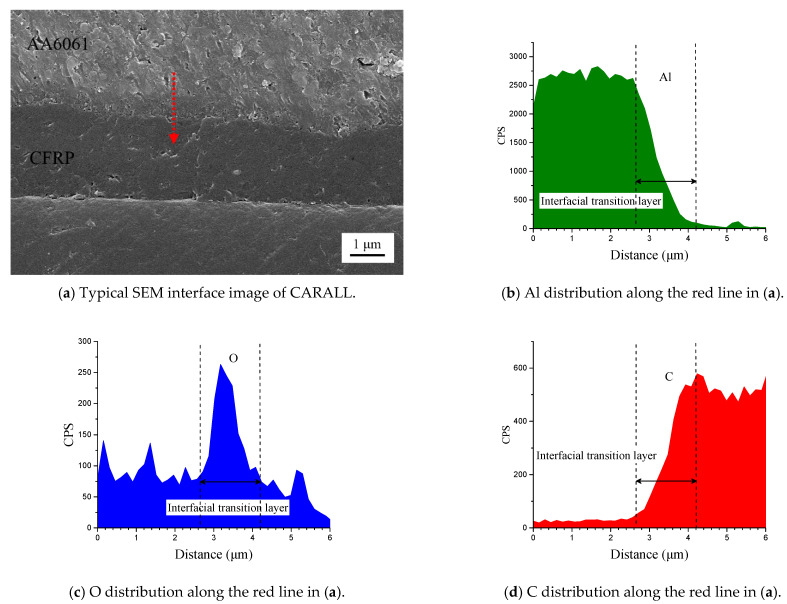
Typical bonding interface of specimen 1387.

**Figure 9 materials-14-01019-f009:**
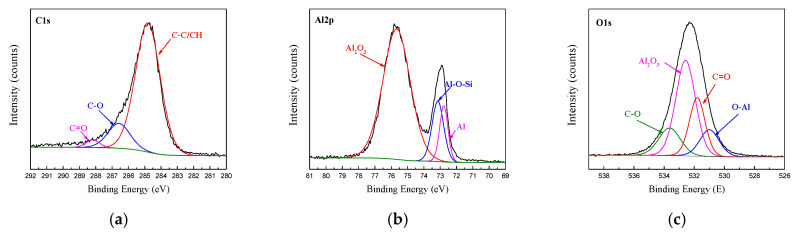
Fitting curves of XPS at the bonding interface of specimens PAA: (**a**) C1s peaks, (**b**) Al2p peaks, (**c**) O1s peaks.

**Figure 10 materials-14-01019-f010:**
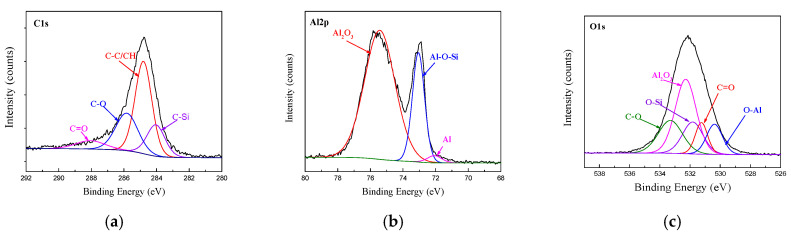
Fitting curves of XPS at the bonding interface of specimens 187: (**a**) C1s peaks, (**b**) Al2p peaks, (**c**) O1s peaks.

**Figure 11 materials-14-01019-f011:**
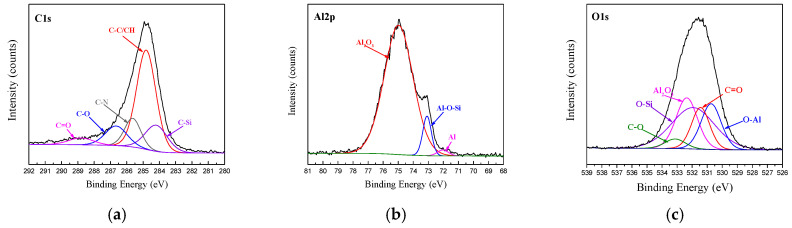
Fitting curves of XPS at the bonding interface of specimens 1387: (**a**) C1s peaks, (**b**) Al2p peaks, (**c**) O1s peaks.

**Figure 12 materials-14-01019-f012:**
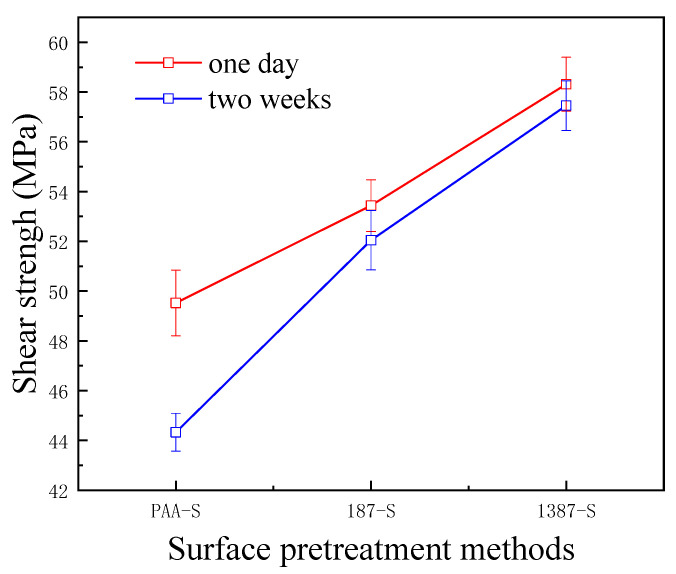
Bonding strength of CARALL for different times.

**Figure 13 materials-14-01019-f013:**
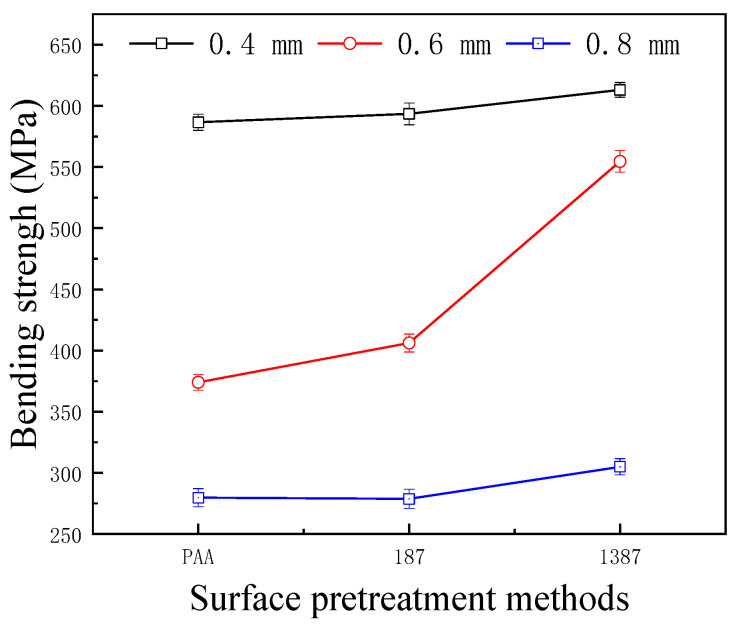
Bending strength of CARALL with different thickness of the AA6061 sheet.

**Figure 14 materials-14-01019-f014:**
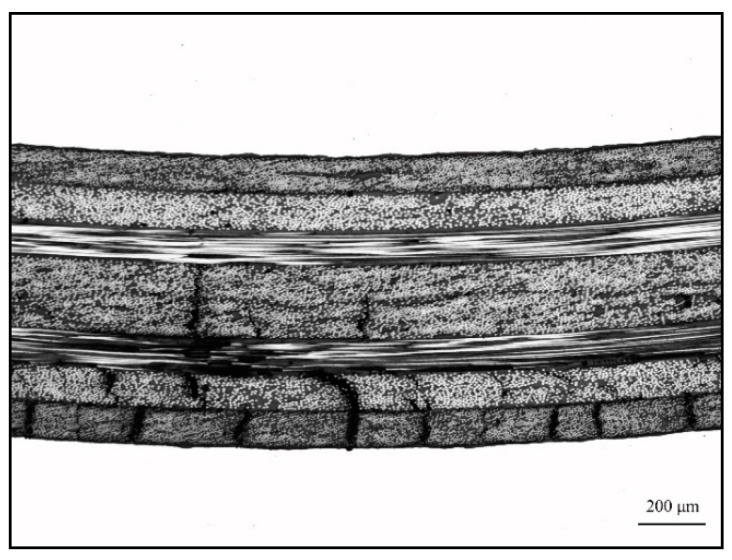
Typical three-point bending interface

**Figure 15 materials-14-01019-f015:**
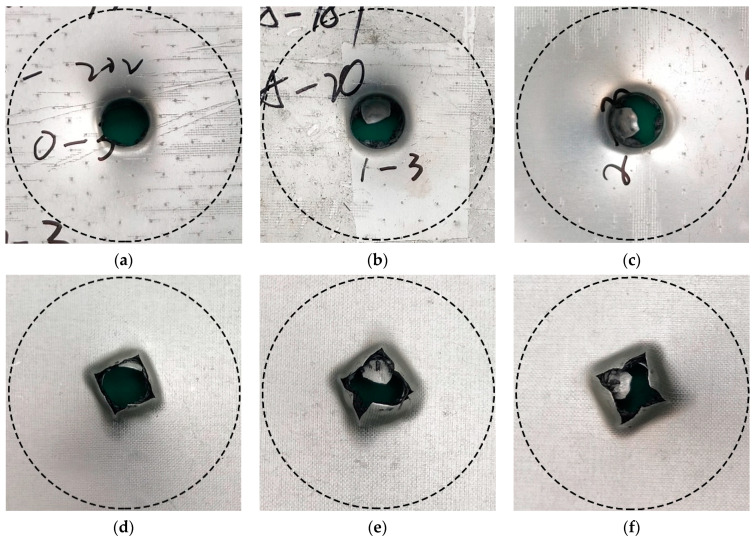
Typical deformation/failure modes of CARALL, ν = 3.42 m/s: (**a**,**d**) are the front and back of PAA-0.6, (**b**,**e**) are the front and back of 187-0.6, (**c**,**f**) are the front and back of 1387-0.6.

**Figure 16 materials-14-01019-f016:**
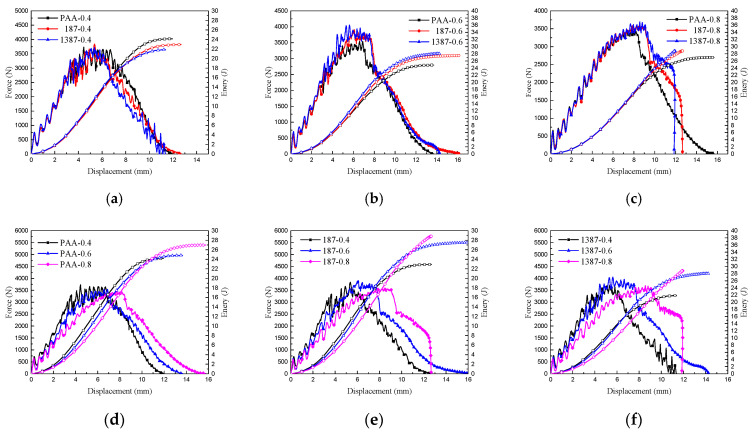
Force–displacement curves and energy absorption of nine types of CARALL: (**a**,**b**,**c**) are the same thickness of AA6061 sheets with different surface treatment conditions, (**d**,**e**,**f**) are different thickness of AA6061 sheets under the same surface treatment condition.

**Table 1 materials-14-01019-t001:** Chemical composition of AA6061.

Materials	Composition (w/%)								
	Si	Fe	Mg	Zn	Mn	Cr	Ti	Cu	Al
AA6061	0.6	0.7	1.2	0.25	0.15	0.2	0.15	0.15	Bal.

**Table 2 materials-14-01019-t002:** Chemical name and structural formula of employed coupling agents.

Silane Coupling Agent	Chemical Name	Structural Formula
A-187	γ-glycidoxy propyl trimethoxy silane	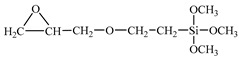
A-1387	Polyamide silane	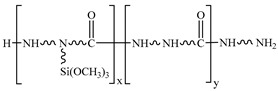

**Table 3 materials-14-01019-t003:** Sequence of surface pretreatments for AA6061 sheets.

Mark	Sequence of Al Surface Pretreatments
PAA	Sanding, surface corrosion, anodizing
187	Sanding, surface corrosion, anodizing, A-187 surface modification
1387	Sanding, surface corrosion, anodizing, A-1387 surface modification

**Table 4 materials-14-01019-t004:** CARALL sample information.

Mark.	Fiber Direction (°)	AA6061 Sheet Thickness (mm)	Surface Pretreatment of AA6061 Sheet	Total Thickness of CARALL (mm)
PAA-0.4	[90/0/45/90/−45/0]_s_	0.4	A	2
PAA-0.6	[0/45/90/−45]_s_	0.6	A	2
PAA-0.8	[0/45/90/−45]	0.8	A	2
PAA-S	[0]_10_	1.0	A	3
187-0.4	[90/0/45/90/−45/0]_s_	0.4	B	2
187-0.6	[0/45/90/−45]_s_	0.6	B	2
187-0.8	[0/45/90−45]	0.8	B	2
187-S	[0]_10_	1.0	B	3
1387-0.4	[90/0/45/90/−45/0]_s_	0.4	C	2
1387-0.6	[0/45/90/−45]_s_	0.6	C	2
1387-0.8	[0/45/90−45]	0.8	C	2
1387-S	[0]_10_	1.0	C	3

**Table 5 materials-14-01019-t005:** Material ratio of CARALL.

Materials	AA6061/CFRP/AA6061
**Loading** **Direction** 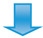			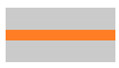
Thickness of AA6061 sheet	0.4 mm	0.6 mm	0.8 mm
Thickness of CFRP	1.2 mm	0.8 mm	0.4 mm
Thickness of CARALL	2 mm	2 mm	2 mm

## Data Availability

The data used to support the findings of this study are available from the corresponding author upon request.

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
