# Peer review of "Effect of Different Coupling Agents on Interfacial Properties of Fibre-Reinforced Aluminum Laminates"

_materials, 2021, doi:10.3390/ma14041019_

Round 1

Reviewer 1 Report

Manuscript is good and I recommends it for publication. But:

Figures numbering should appear in the text before the individual Figures are placed.
2. The numbering of the patterns makes reading difficult the patterns
3. Formula 5 is too basic and is not new, to appear in a research paper
4. There are no markings and descriptions in the figure caption and in the Figures. The best example is Figure 14

Title:

The title should be more specific. It should present name of coupling agents that influence on the properties. It is quite general to mention that "different coupling agents" are being analyzed. It is not a big mistake, but in my opinion the title could be improved. However, it is not necessary.

Abstract:

  1. The abbreviations MCI, FML and CARALL should be expanded with words starting with capital letters. This will be brighter for the reader
  2. The rest of the abstrat is correct and I have no comments to it.

Introduction:

  1. Figure 1 appears in the manuscipt before the reference in the text. There should be information in the text first, then put a Figure.
  2. Is the Figure 1 made by the authors? If not, please add references.
  3. The authors cite and briefly describe the achievements of other researchers, which is very well perceived
  4. It is worth adding a reference regarding the influence of wettability on bond strength. The authors report that several studies have been conducted, but do not cite them. One or two references would certainly make this information credible
  5. Despite these comments, I think the Intruduction section is well prepared.

Experimental Procedures:

  1. Have the authors checked the solutions without conducting ultrasonic oscillations? Does it have a big impact on the final result?
  2. Line 140: The room temperature ranges from 15 ° C to 25 ° C. Please specify the temprature more precisely.
  3. Figure 2: no caption under the figure for (a) (b) (c)
  4. Formula (5) is too obvious to be included in a research paper
  5. Section 2.5. Characterization should change the name. Maybe "Structure and chemical composition"? Characterization is too general.

Results and discussion

  1. Fig 5. Please provide the original SEM image along with the photo taking parameters
  2. The authors write about wettability, but did not conduct wettability studies. In my opinion, the bonding can be related to wettability, but it cannot be determined on this basis.
  3. Please divide Figure 9 into (a) (b) (c) (d). Similar remark to Figures 10 and 11
  4. Table 4. Is the word "Punching" appropriate?
  5. Overall the results are described clearly. You could refer to the work of other researchers more often and compare them with your results. However, it is not required.

Conclusions:

  1. The conclusions are logical and relate to the conducted research.

References:

  1. The authors chose the literature well. 35 references is enough. I have not detected any citations not related to the topic.

Reviewer 2 Report

The paper is well written and clearly shows the experimental approach with sufficient details.

I suggest the authors include in the reference also more recent literature results to emphasize the originality of their work better. In particular, much literature exists about CARALL and fiber metal laminates.

Besides the characterization of materials properties should consider also repeatability and confidence interval of the experimental results obtained (e.g. Figure 13).

Reviewer 3 Report

The paper present the results obtained on Aluminium alloy-CFRP laminates (CARALL) where the Al sheets have been submitted to 3 diferent types of preparation in order to increase the adherence and mechanical properties. 

The paper can be published, after some revisions.

1) There are many abbreviations and not all are explained as appearing in the text (e.g. CFRP, line 25) or are never explained (e.g. BGM, line 76). Other are not necessary, their use making the text difficult to read, like e.g. UD for unidirectional or MCI for metal composite interface, which can be easily replaced by "interface" avoiding also errors (like e.g. MIC, line 85). 

2) Also Table 3 appear twice, should be renumbered. 

3) In the second Table 3, "CARALL sample information: the lay -up method is not described, it should be pointing to the significance of the different lay-ups for the tests performed. 

4) Figures 6-8: what is the unit GPS? Should not be CPS (counts per second)? Also how was the interface thickness defined? Does it depends on treatment and how? In the SEM picture of Fig. 8 it seems to me that there is a fiber connected to Al. Is this the reason for the different scale values in the maps? And why was not Si content also analyzed (for B and C) as in the XPS spectra? 

5) Mechanical tests: the paper should provide a better material and a reader will expect to see in the plots (figs 12, 13 and 15) the improvements. Please include some reference material from literature (e.g. Al alloy and/or laminates without the surface treatments.

6) Figure 12, why "after 2 weeks"? What is the relevance for the material life span? I think the important parameter is here a curve extrapolation to see what really happens after a relevant period. 

7) In figure 13 the error bars are missing, please include them. Also some pictures (microscopy or SEM) for the samples after the bending test will be nice. 

With these amendments I expect the paper to be suited for publication.

Reviewer 4 Report

This manuscript is very poorly written. Some figures are incorrectly and inconsistently described. The authors describe Fig. 16 that does not exist in the manuscript. Sentences are meaninglessly duplicated or placed in wrong places. In addition, test results are not fully evaluated to investigate the effect of MCI. 

Round 2

Reviewer 4 Report

The manuscript still needs to be improved further in many places. "Fig 14" mentioned in line 337 on page 15 is not related to any figure showing the bending process. The caption of Figure 14 should be revised to correctly match the front and back images of each case. There are two "Table 4" in the manuscript. In addition, there are too many typos and errors. For example, "Puerto Rico, USA" in line 102 on page 3, "~line in (c)" in line 223 on page 10, "that he anodizing" in line 273 on page 13 and so on. The authors needs to read the manuscript thoroughly again to find and correct them. 
